# Saltatory axonal conduction in the avian retina

Christoph T. Block, Malte T. Ahlers [ID], Christian Puller [ID], Max Manackin, Dipti R. Pradhan and Martin Greschner

*Visual Neuroscience, Department of Neuroscience, Carl von Ossietzky University Oldenburg, Oldenburg, Germany*

Handling Editors: Nathan Schoppa & Jonathan B. Demb

The peer review history is available in the Supporting Information section of this article (https://doi.org/10.1113/JP288664#support-information-section).

Christoph T. Block studied neuroscience and focuses on the functional aspects of retinal ganglion cells with a methodological focus on data science, from raw data handling to high-level analysis, often including the use of machine learning to identify patterns. Malte T. Ahlers studied biology and philosophy at the University of Oldenburg. In his PhD research he investigated thermal and non-thermal effects of high frequency electromagnetic fields on retinal electrophysiology. After working as a freelance hardware and software developer and co-founding a robotics start-up he joined the Greschner lab as senior scientist. Christian Puller studied biology and works as a senior scientist focusing on the structure and function of retinal circuitry in vertebrates. He investigated the anatomy of the outer mammalian retina during his PhD, studied the physiology of retinal output neurons as a postdoc, and today analyses connectomics of retinal circuitries in large EM datasets.

C. T. Block, M. T. Ahlers and C. Puller contributed equally to this work.

This article was first published as a preprint. Block CT, Ahlers MT, Puller C, Manackin M, Pradhan DR, Greschner M. 2022. Saltatory axonal conduction in the avian retina. bioRxiv. https://doi.org/10.1101/2022.07.04.498722

**Abstract figure legend** Intraretinal axonal spike conduction was studied across multiple avian and mammalian species using high-resolution multielectrode arrays. On the electrode array, saltatory axonal spike conduction was readily apparent and distinguishable from non-saltatory conduction. The highest conduction velocities were observed exclusively in saltatory axons. However, slow saltatory axons exhibited a surprisingly large overlap in conduction velocities with unmyelinated axons. Intraretinal axonal conduction velocities observed in birds were up to four times faster than in rodents.

**Abstract**  In contrast to most parts of the vertebrate nervous system, ganglion cell axons in the retina typically lack myelination. In the majority of species, ganglion cell axons only become myelinated after leaving the retina to form the optic nerve. The avian retina, however, presents a remarkable exception in that ganglion cell axons are partly myelinated in the retinal nerve fibre layer. It was hypothesized that the optically detrimental properties of retinal myelination are evolutionarily offset by advantages in spike conduction velocity. Using high-resolution multielectrode array recordings, we analysed the spike conduction in the retina of various avian species in comparison to mammalian species. Indeed, mammals showed lower conduction velocities than avian species. Myelinated axons typically achieved higher conduction velocities than unmyelinated axons. Notably, some myelinated axons exhibited conduction velocities lower than those of unmyelinated axons. Anatomical analyses revealed that myelination in the nerve fibre layer was accompanied by the formation of nodes of Ranvier. The internode length was positively correlated with the axon diameter. In physiological recordings, the spatial extent of simultaneously active nodes was positively correlated with the conduction velocity. Conversely, the internode length and the activation kinetics of a node were weak predictors of conduction velocity. Overall, this study illuminates the unique features of the avian retina and offers insights into the functional requirements and evolutionary pressures of myelination affecting conduction velocity in the nervous system.

(Received 30 January 2025; accepted after revision 25 July 2025; first published online 22 August 2025)

**Corresponding author** M. T. Ahlers: Visual Neuroscience, Department of Neuroscience, Carl von Ossietzky University Oldenburg, Oldenburg, Germany.　　Email: malte.ahlers@uol.de

## Key points

- Intraretinal saltatory axonal spike conduction was studied across multiple avian species using high-resolution multielectrode arrays.
- The highest conduction velocities were observed exclusively in saltatory axons, while the lowest were found in non-saltatory axons. However, slow myelinated axons exist, exhibiting a surprisingly large overlap in conduction velocities with unmyelinated axons.
- The spatial extent of a spike showed a strong positive correlation with conduction velocity.
- The internodal length exhibited a positive correlation with axon diameter, and the variability in internodal length was smaller within individual axons than across axons.
- Intraretinal axonal conduction velocities across species appear to align with their ecological niche. The maximal intraretinal spike conduction velocity observed in birds was up to four times faster than in rodents.

## Introduction

Myelination is a prevalent feature of vertebrate axons, enabling fast saltatory spike conduction. Myelin-shielded sections of the axon are periodically interrupted by non-myelinated nodes enriched with voltage-gated ion channels exposed to the extracellular space. Myelination decreases membrane capacitance and increases membrane resistance; as a result it increases the membrane's length constant and decreases its time constant. This allows an activated node to depolarize subsequent nodes over a longer distance, avoiding time- and energy-consuming active spike generation over the entirety of the axon, greatly increasing conduction velocity.

Myelin is a protein- and lipid-rich optically dense material exhibiting strong light scattering in aqueous

media (Blanke et al., 2023; DePaoli et al., 2020). It potentially compromises the optical properties of the retinal nerve fibre layer, through which light must travel to reach the photoreceptors. In humans, a rare pathological myelination of retinal axons (myelinated retinal nerve fibres, MRNF) results in locally impaired visual acuity (Grzybowski & Winiarczyk, 2015; Lee et al., 2015). In the healthy mammal, retinal ganglion cells (RGCs) remain unmyelinated until they pass the lamina cribrosa, after which they form the optic nerve and become myelinated (Fujita et al., 2000). Accordingly, the mammalian retina typically lacks any oligodendrocytes, the glial cells responsible for forming the myelin sheaths around axons in other parts of the central nervous system (Gao et al., 2006). A notable exception among mammals is the rabbit, where intraretinal myelination by oligodendrocytes is present, though limited to a region above the visual streak (Hughes, 1971; Schnitzer, 1985; Vaney, 1980). Intraretinal myelination has also been observed in fish, specifically in presumed centrifugal axons (Witkovsky, 1971; Wolburg, 1980).

In contrast, myelinated ganglion cell axons are commonly found in the avian retina (Imagawa et al., 1999; Inoue et al., 1980; Nakazawa et al., 1993; Seo et al., 2001; Won et al., 2000). The percentage of myelinated axons varies across retinal regions (Imagawa et al., 1999). All centrifugal axons were described to be myelinated intraretinally (Wilson & Lindstrom, 2011). A distinguishing feature of myelin in avian retinas is its loose ultrastructure, characterized by largely missing intraperiod lines and proteolipid proteins, major attributes of compact myelin in the central nervous system (Inoue et al., 1980; Kohsaka et al., 1980). In the avian retina, Müller cells, rather than oligodendrocytes, are the primary myelinating cells (Seo et al., 2001; Won et al., 2000), Only compact myelin is found in the optic nerve (Fujita et al., 2001). The myelination of RGC axons in the nerve fibre layer is biased toward thicker axons (Nakazawa et al., 1993).

From an evolutionary perspective, the detrimental optical consequences of intraretinal myelination should be counterbalanced by a significant improvement in spike conduction velocity or advantages such as energy efficiency or temporal precision. To examine this hypothesis, we studied intraretinal spike conduction velocities in species with and without myelinated axons and with behaviours indicating potentially different needs for visual processing speeds. Specifically, we recorded the electrical activity of RGC axons with high-resolution multielectrode arrays (MEAs) of rodents and birds and estimated the respective spike conduction velocities. Additionally, we functionally and anatomically analysed the saltatory spike conduction in the nerve fibre layer of the avian retina.

## Methods

### Ethical approval

All experiments were performed in compliance with the German Animal Protection Law (Tierschutzgesetz), as issued by the German Federal Government, and the guidelines of the European Union for the use of animals in research. The procedures were approved by the animal welfare officer of the Carl von Ossietzky University of Oldenburg (§4 animal protocol) and carried out in compliance with the AVMA Guidelines for Euthanasia (Leary et al., 2020). Animal breeding and housing followed the standards given in Directive EU 2010/63 and was approved by the local authorities (LAVES Niedersächsisches Landesamt für Verbraucherschutz und Lebensmittelsicherheit). The procedures complied with *The Journal of Physiology*'s checklist provided to authors (Grundy, 2015; O'Halloran, 2024).

### Animals

Young pigeons (*Columba livia* f. *domestica*) and adult guinea pigs (*Cavia porcellus*) were terminally anaesthetized by pentobarbital (Narcoren, Boehringer GmbH, Ingelheim, Germany). Young European quails (common quail, *Coturnix coturnix*), young Bankiva chicken (red junglefowl, *Gallus gallus*) and adult zebra finches (*Taeniopygia guttata castanotis*) were decapitated. Adult C57BL/6J mice were killed by cervical dislocation. Animals of both sexes were included in the study, and no sex-specific differences were readily apparent. All animals were housed under a natural light–dark cycle, and experiments were performed during daylight hours. Access to water and food was provided *ad libitum*.

### MEA recordings and analysis

Avian retinas were dissected in Ringer solution (100 mM NaCl, 6 mM KCl, 1 mM $CaCl_2$, 2 mM $MgSO_4$, 1 mM $NaH_2PO_4$, 30 mM $NaHCO_3$, 50 mM glucose), bubbled with carbogen (95% $O_2$ and 5% $CO_2$) (Stett et al., 2000). Mammalian retinas were dissected in carbogenated Ames' medium (Sigma-Aldrich GmbH, Taufkirchen, Germany). Dissections were carried out under infrared illumination. Recordings from mammalian retinas were conducted in Ames' medium at 36.5°C in the chamber. Recordings from avian retinas were conducted in Ringer solution at 37.5°C in the bath chamber if not indicated otherwise (Fig. 6*E–H*). In most cases, the recording temperature was below the physiological body temperature throughout the daily cycle of the respective animal (mouse: 34.5–38°C (van der Vinne et al., 2020), 35.8–38.7°C (Yasumoto et al., 2015); guinea pig: 37.4–38°C (Kainulainen et al., 2018),

37.6°C–38.4°C (Thorne et al., 1987); chicken: 39–41.3°C (Shimmura et al., 2015), 41.1–41.7°C (Savory et al., 2006); quail: 41–42°C (Laurila et al., 2005), 39.9–42.6°C (Underwood et al., 1999); zebra finch: 39.7–41.8°C (Zagkle et al., 2020), 39.2–41.4°C (Sköld-Chiriac et al., 2015); pigeon: 39–41.6°C (Rashotte et al., 1995), 39.3–42°C (Rashotte et al., 1999)). In general, birds maintain a higher body temperature than mammals, and also show greater body temperature fluctuations depending on the animals' activity (reviewed in Prinzinger et al., 1991). For recording, a small piece of peripheral retina was mounted with ganglion cell side down onto the MEA. Most recordings were performed with a 2.6 × 2.6 mm$^2$ complementary metal-oxide-semiconductor (CMOS) MEA with an electrode pitch of 42 μm (3Brain AG, Pfäffikon, Switzerland) that allowed the recording of axons over long distances in sufficient numbers. We conducted a limited number of recordings at a higher spatial resolution with a 1 × 1 mm$^2$, 16 μm pitch CMOS MEA (Multichannel Systems GmbH, Reutlingen, Germany) (Fig. 6*E–H*). Sampling rates of the two systems were 18 and 20 kHz, respectively. Retinas were light stimulated at a low photopic light level.

Recordings were analysed offline to isolate the spikes of different cells. After bandpass filtering (60 Hz–4 kHz), candidate spike events were detected using a threshold on each electrode. The voltage waveforms on the electrode and neighbouring electrodes around the time of the spike were extracted. Clusters of similar spike waveforms were identified as candidate neurons if they exhibited a refractory period. Duplicate recordings of the same cell were identified by temporal cross-correlation and matching electrical images and removed.

After spike sorting, the spatio-temporal spike-triggered average electrical signal was calculated for each cell (Bucci et al., 2024; Greschner et al., 2014; Litke et al., 2004; Radivojevic et al., 2017; Seifert et al., 2023; Zeck et al., 2011). Specifically, for each electrode, the voltage waveform during the time period from −1.62 ms to 4.42 ms around each spike was extracted. All spike waveforms were upsampled to 200 kHz using cubic spline interpolation, aligned to the somatic voltage minimum of the spike, and averaged. All further analyses worked with this average electrical signal. To construct the time-collapsed two-dimensional electrical image, the minimum value of the averaged waveform at each electrode was displayed according to the spatial layout of the MEA. If indicated in the figure legend, the signals on the soma electrodes were saturated to improve the visual representation of the axon. RGCs with axons of insufficient length were excluded. The analysis was restricted to centripetal axons originating from RGCs, as confirmed by the direction of signal propagation relative to the cell body observed in the electrical image.

The conduction velocity of a spike was estimated by linear regression of the Euclidean distances between the identified spatial positions on the axon and the corresponding minima in the time course of the signal waveforms. Axons were traced manually in the electrical images based on local minima. All further analyses were based on these axon positions. The sampling rate of the multielectrode recording, combined with the electrode-to-electrode distance, limits the detectable range of conduction velocities. However, by tracking spike propagation across the MEA and defining the conduction velocity as the linear regression of all sample points, we ensured that the detectable range of conduction velocities far exceeded the biological range observed. The distance measurement was done in the projected spatial electrical image and ignored potential depth variations of the axons in the nerve fibre layer. This projection biased the measured axonal distances to slightly smaller values. We estimated its influence anatomically by tracing NF200-positive axons (see 'Immunostaining and light microscopy' below) and found that the conduction velocity was possibly underestimated by 0.05 ± 0.02% ($n = 20$), ignoring possible artefacts of the anatomical procedure.

To distinguish saltatory and non-saltatory axons (Fig. 2), a continuous spatio-temporal signal profile along the axon was extracted. First, the spatial path was defined as each electrode along a linear connection of the manually traced axon. The spike time for each electrode on this path was estimated from the fitted velocity. The distinction between saltatory and non-saltatory axons was determined using a continuity index, which quantified the smoothness of these signal profiles. For each axon, the continuity index was calculated as the ratio of the robust means of local maxima and local minima along the signal profile. To ensure the continuity index was in the range between 0 and 1, the global maximum of the entire dataset was subtracted prior to forming the ratio. A low value of the continuity index represents a more discontinuous saltatory axonal signal profile, while a high value represents a more continuous non-saltatory signal profile. A Gaussian mixture model was applied to the continuity indices and the conduction velocities of all traced axons. Based on this model, axons were categorized into two groups of saltatory and non-saltatory spike conduction, respectively. RGCs with a likelihood below 0.90 of belonging to either cluster were excluded from further analysis. After categorization, several parameters were estimated for the saltatory axons interpreting the manually defined local minima as the positions of nodes of Ranvier. When appropriate, corresponding measurements on non-saltatory axons were based on their local minima.

The spatial extent of a spike was defined as the average extent of its negative signal along the axon (Fig. 3). Specifically, we sequentially examined all nodes along an

axon. At the time of the strongest negative deflection at a given node, that is, when the spike was at that node, we extracted the signal at all nodes, aligned the signals by their distance along the axon relative to the current spike position, and averaged the signals. We then estimated the distance between the nearest antidromic and orthodromic voltage zero crossings. Cells without a sufficient number of measurable nodes were excluded. The node amplitude was reported as the robust mean of the minimum signals across all nodes of each axon (Fig. 4*A*–*D*). Basing this amplitude measurement only on nodes with large amplitudes did not change the overall correlation results. The onset duration of the spike was defined as the robust mean of the time from half-minimum to minimum of the signal waveform at each node (Fig. 4*E*–*H*). The offset duration was defined accordingly. As all measurements were performed on the spike-triggered average electrical signal, they are potentially influenced by the precision of spike time alignment. In particular, variability in spike velocity introduces jitter, as spikes are aligned to the time of the somatic spike. Lower alignment precision could potentially lead to a broader spike waveform with reduced amplitude.

The maximal axonal lengths from the dorsal retinal rim to the optic nerve head for the studied species were assumed to be: pigeon: ∼15 mm (Marshall et al., 1973); zebra finch: ∼6.5 mm (Liu et al., 2023); guinea pig: ∼8 mm (Howlett & McFadden, 2007); and mouse: ∼3 mm (Duda et al., 2025). For pigeon and zebra finch, the optic nerve head was assumed to be offset by ∼45 degrees (Marshall et al., 1973). Estimates for the pigeon assumed a flight velocity of 100 km/h.

### Immunostaining and light microscopy

For immunostaining, the tissue was fixed in 4% paraformaldehyde in 0.01 M phosphate-buffered saline (PBS, pH 7.4) for 20–30 min at room temperature. After fixation, the tissue was cryoprotected overnight with 30% sucrose in PBS and stored at −20°C until use.

Immunostaining was performed by an indirect fluorescence method. Retinal wholemounts were incubated at room temperature for 2–3 days in the primary antibody solution containing 5% normal donkey serum, 1% bovine serum albumin, 1% Triton X-100 and 0.02% sodium azide dissolved in PBS. Primary antibodies used in this study were anti-myelin basic protein (MBP, rat, monoclonal, clone 12, Bio-Rad Laboratories, Hercules, CA, USA, cat. no. MCA409S, 1:1000), anti-Neurofilament 200 (NF200, rabbit, polyclonal, Sigma-Aldrich, cat. no. N4142, 1:1000), and anti-voltage gated sodium channel, pan (panNa$_v$, mouse, monoclonal, clone K58/35, Sigma-Aldrich, cat. no. S8809, 1:1000). Secondary donkey antibodies (1:500, conjugated to Alexa 488, 568, 647, Thermo Fisher Scientific, Waltham, MA, USA, or Cy3, Jackson ImmunoResearch Laboratories, West Grove, PA, USA) were incubated at room temperature for 4 h in the same incubation solution. The tissue was mounted on glass slides and coverslipped with Vectashield (Vector Laboratories, Burlingame, CA, USA). Spacers between glass slides and coverslips were used to avoid squeezing of the tissue.

Overview image stacks were taken with a confocal laser scanning microscope (Leica TCS SL, Leica Microsystems GmbH, Wetzlar, Germany), equipped with a ×40/NA 1.25 oil immersion objective. For high resolution images of panNa$_v$ clusters and corresponding size measurements, image stacks were acquired with a ×63/NA 1.32 oil immersion objective. All confocal stacks were acquired with z-axis increments of 0.2 μm. For measurements of internode lengths and axon diameters, tile scans of image stacks along axons were performed with a Leica DM6B fluorescence microscope, equipped with a motorized stage in combination with a ×40/NA 1.3 oil immersion objective. Automated acquisition and stitching of individual tiles was performed in the Leica LASX software package. The segmented line tool in Fiji was used to measure the distance between panNa$_v$ clusters, that is nodes of Ranvier, along individual NF200-positive axons. Internodal lengths were measured as the distance between the centres of panNa$_v$-positive clusters. Note that the used definition of internode length for the anatomical measurements includes the para- and juxtaparanodes and refers to the full distance between panNa$_v$ clusters. Each internode length measurement was accompanied by a measurement of axon diameter as an orthogonal line spanning the neurofilament-positive area at the approximate halfway point between a given pair of clusters. For the measurements of panNa$_v$ cluster lengths and diameters, a single image plane was manually selected per cluster to include the largest diameter of the whole tubular structure. A region of interest was thresholded using Otsu's method. Individual lengths values represent the mean lengths averaged across measurements of both sidewalls of each cluster. Maximum intensity sideview- or z-projections are shown in all figure panels of microscopic images and were done in Fiji. The brightness and contrast of the final images were adjusted in Adobe Photoshop.

### Statistical analyses

All tests and analyses were performed in MATLAB (MathWorks, Inc., Natick, MA, USA). In the Results section, measurements are reported as the mean ± SD. The robust mean was defined as the mean of the central 80% of the data. No further outlier selection was performed. The lower and upper bounds of the 95% confidence interval are reported for the correlation

coefficient. Tests for zero correlation of the investigated parameters (number of active nodes, node amplitude, node onset duration) with the conduction velocity were rejected in all cases. All tests for differences between distributions were performed using Wilcoxon's rank sum tests and rejected equal medians in all cases. For the analysis of internode length variation within and across axons (Fig. 6*B*), the quartile coefficient of dispersion was used (Bonett, 2006). The coefficient of dispersion across axons was bootstrapped with replacements.

## Results

### Electrical images reveal saltatory spike conduction

We employed large-scale MEA recordings to study the influence of intraretinal axonal myelination on spike conduction. The high electrode density of these arrays allowed for the detailed analysis of the spatiotemporal electrical activity of the recorded neurons (Bucci et al., 2024; Greschner et al., 2014; Litke et al., 2004; Radivojevic et al., 2017; Seifert et al., 2023; Zeck et al., 2011). The electrode-wise, spike-triggered average electrical activity provides insight into how a spike travelled along the axon of a RGC. The spatial structure was highlighted by collapsing the time dimension using a minimum projection of the electrical activity of the recorded cell (electrical image, Fig. 1). As expected, a high-amplitude somatic area, a surrounding dendritic area and an axon extending toward the optic disk were distinguishable.

Many recorded RGCs in the European quail showed a continuous axonal spike conduction (Fig. 1*A–C*), as is common for RGC axons of most mammalian species. However, a large fraction of the recorded

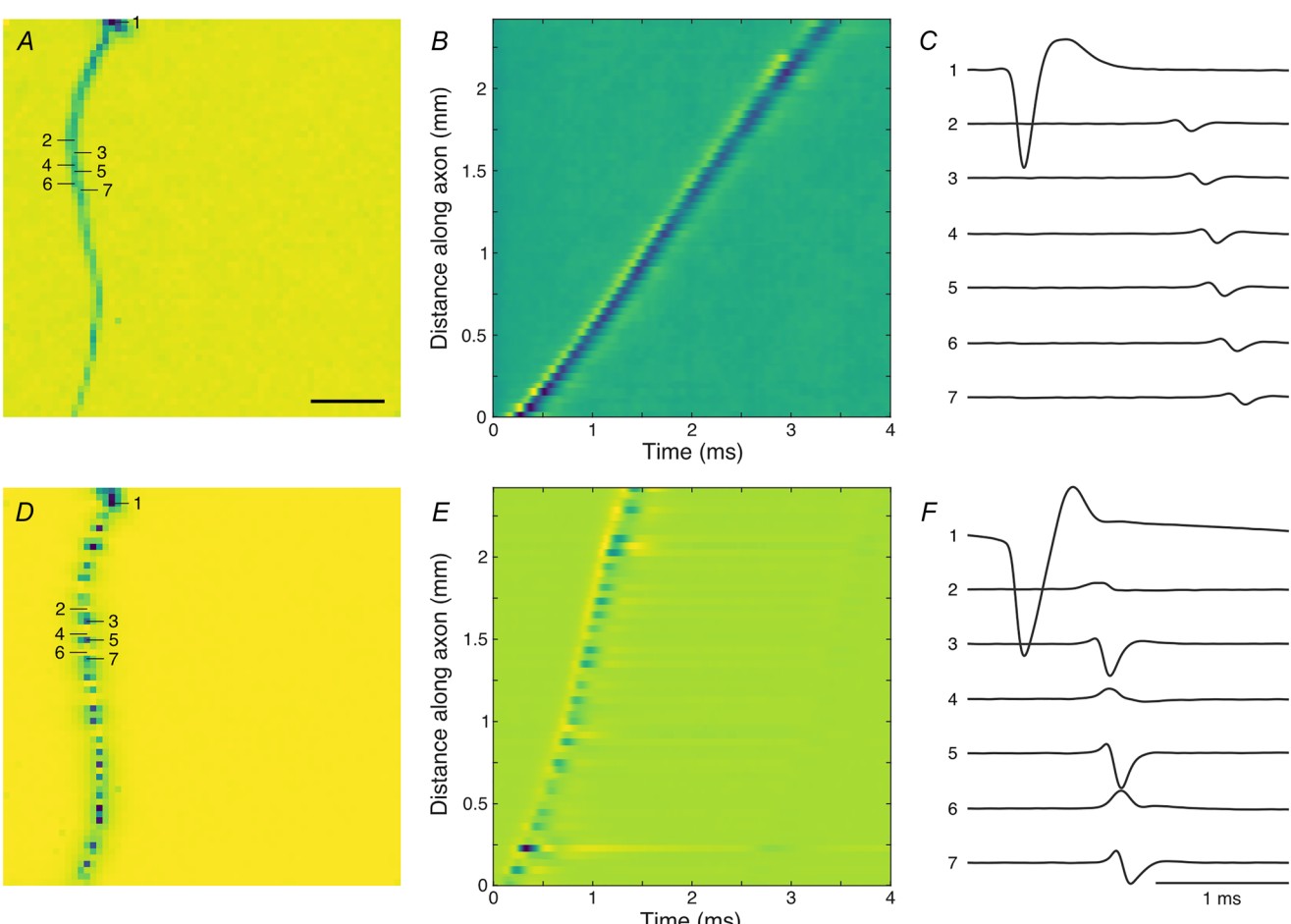

**Figure 1. The electrical images of European quail retinal ganglion cells revealed axons with and without saltatory conduction**

*A*, electrical image of non-saltatory RGC with conduction velocity $v = 0.79$ m/s. Minimum voltage projection along the time dimension of the spike-triggered average electrical activity. Cold colours indicate negative potentials. Soma amplitude saturated for visual clarity. $64 \times 64$ electrodes, scale bar: 500 μm. *B*, signal waveform along the axon. Cold colours indicate negative potentials. *C*, signal waveform for seven selected electrodes indicated in *A*. Scale bar: 1 ms. *y*-Axis in arbitrary units, identical *y*-scale in *C* and *F*. *D–F*, as *A–C* for saltatory RGC with $v = 2.02$ m/s.

quail RGCs displayed discontinuous, that is saltatory, axonal conduction (Fig. 1*D–F*). In these cases, we observed distinct axonal regions of high-amplitude signals alternating with low-amplitude regions of opposite signal polarity, as expected for myelinated axons with passively conducting segments, periodically interrupted by active node-of-Ranvier-like structures (Fig. 1*D* and *F*). Analysis of the spike conduction along the axon over time revealed a roughly linear distance–time relation in both cases (Fig. 1*B* and *E*). The slope of a linear fit of the times of maximum negative deflection and the corresponding distances along the axon was used as an estimate of the axonal conduction velocity.

We quantified the degree of signal continuity of each axon by calculating the ratio between the local maxima and minima along its trace in the electrical image. The distribution of this continuity index was bimodal (Fig. 2), reflecting saltatory and non-saltatory conducting axons. For further analysis, we excluded neurons with a low likelihood of belonging to either group (Fig. 2, grey cluster). As expected, saltatory axons displayed overall higher conduction velocities (saltatory: $1.25 \pm 0.35$ m/s; $n = 337$; non-saltatory: $0.71 \pm 0.12$ m/s; $n = 163$; rank sum test $P < 0.001$). While high conduction velocities were only found in saltatory axons, surprisingly, lower conduction velocities were present in both non-saltatory and saltatory axons.

### Large variability of biophysical parameters

Various biophysical parameters determine the velocity of saltatory spike conduction. The analyses of electrical images allowed the estimation of different relevant parameters of intraretinal saltatory conduction in the avian retina over hundreds of cells. Foremost, the signal spread beyond an active node can be expected to play a dominant role in the axonal conduction velocity. We measured the extent of active nodes, that is nodes simultaneously active (Fig. 3*A–C*). These indicate how far the signal on the node with currently the largest signal amplitude is spreading. The total extent of simultaneously active saltatoric nodes was on average $311 \pm 114$ µm ($n = 335$; observed maximum: 722 µm) in the quail retina and indeed correlated with the conduction velocity (Fig. 3*D*; correlation coefficient: 0.73, CI: 0.67, 0.77; $P < 0.001$). The average signal extent on a non-saltatory axon was lower ($161 \pm 44$ µm; $n = 161$; rank sum test: $P < 0.001$) and hardly correlated with the conduction velocity (correlation coefficient: 0.37, CI: 0.23, 0.50; $P < 0.001$). The extent of active antidromic nodes, that is the nodes still negative after activation, was correlated with the extent of active orthodromic nodes (correlation coefficient: 0.58, CI: 0.51, 0.65; $P < 0.001$), though the antidromic extent exceeded the orthodromic extent by a factor of $1.97 \pm 1.07$ (rank sum test: $P < 0.001$).

The extent of orthodromically active nodes is determined by several parameters, such as the length constant of the axon, which indicates the extent of the signal spread, the overall sensitivity of the nodes, and their activation kinetics. While none of these factors were accessible directly in our recordings, the amplitude and the onset time of the electrical signals can serve as proxies. The average nodal amplitude (Fig. 4*A–D*) exceeded that of average non-saltatory axons by a factor of $2.33 \pm 1.18$. In non-saltatory axons, the axonal amplitude roughly scaled with the conduction velocity (Fig. 4*D*; correlation coefficient: $-0.47$, CI: $-0.58$, $-0.34$; $P < 0.001$). However, for saltatory axons, this was not apparent (correlation coefficient: $-0.24$, CI: $-0.34$, $-0.14$; $P < 0.001$). Additionally, the signal onset of a given node (Fig. 4*E–H*) showed no apparent correlation with conduction velocity (Fig. 4*H*; correlation coefficient: $-0.16$, CI: $-0.26$, $-0.06$; $P = 0.003$). The onset and offset times of the signal were strongly correlated (correlation coefficient: 0.81, CI: 0.76, 0.84; $P < 0.001$).

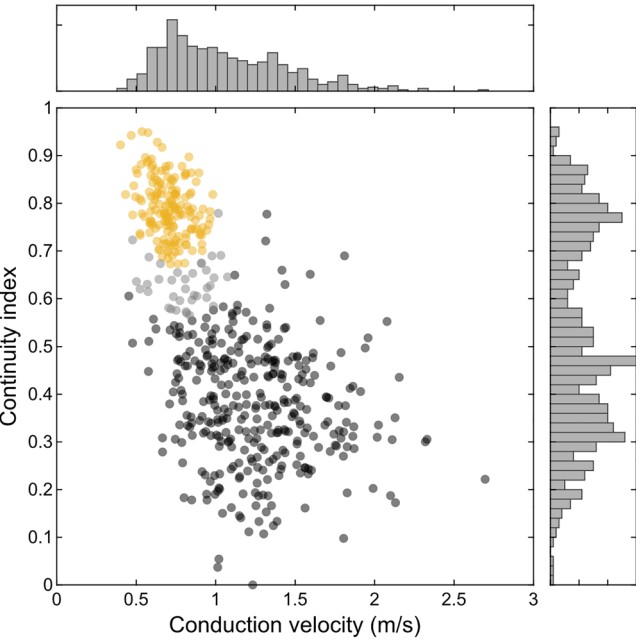

**Figure 2. Saltatory and non-saltatory axons exhibited different conduction velocities**

RGCs with sufficient axon length from one European quail retina were classified into 337 saltatory (black) and 163 non-saltatory (yellow) axons. The overlap region of 36 axons (grey) was excluded from further analyses (likelihood <0.90, mixture of Gaussians). The continuity index is the robust mean peak amplitudes divided by the robust mean trough amplitudes along the axon. See Methods for details. Figures 3 and 4 present data from the same cells.

## Voltage-gated sodium channels are clustered at nodes

Next, we anatomically analysed the myelination pattern and the sodium channel distribution in the temporal region of a quail retina (Fig. 5A–C). Sodium channels clustered in short, tubular structures. These clusters were free of myelin. Unmyelinated axons showed no sodium channel clusters and only diffuse panNa$_v$ immuno-reactivity which rarely exceeded background levels.

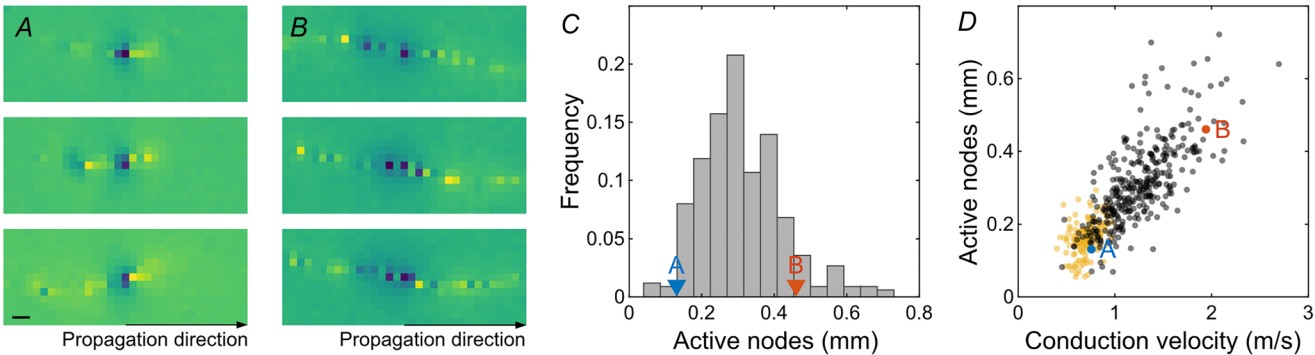

**Figure 3. Spatial extent of active nodes was correlated with conduction velocity in the European quail retina**

*A*, enlarged view of the spatial electrical activity of three example nodes from one RGC axon. Electrical images were aligned in space and time to the local minimum of the respective node. Blue colours indicate negative values. Maximum amplitude saturated for visual clarity. Scale bar: 100 µm. *B*, as *A* for a second RGC with a high number of simultaneously active nodes. *C*, histogram of average simultaneously active nodes. RGCs of *A* and *B* marked. *D*, spatial extent of active nodes against conduction velocity for saltatory (black) and non-saltatory (yellow) axons. RGCs of *A* and *B* marked.

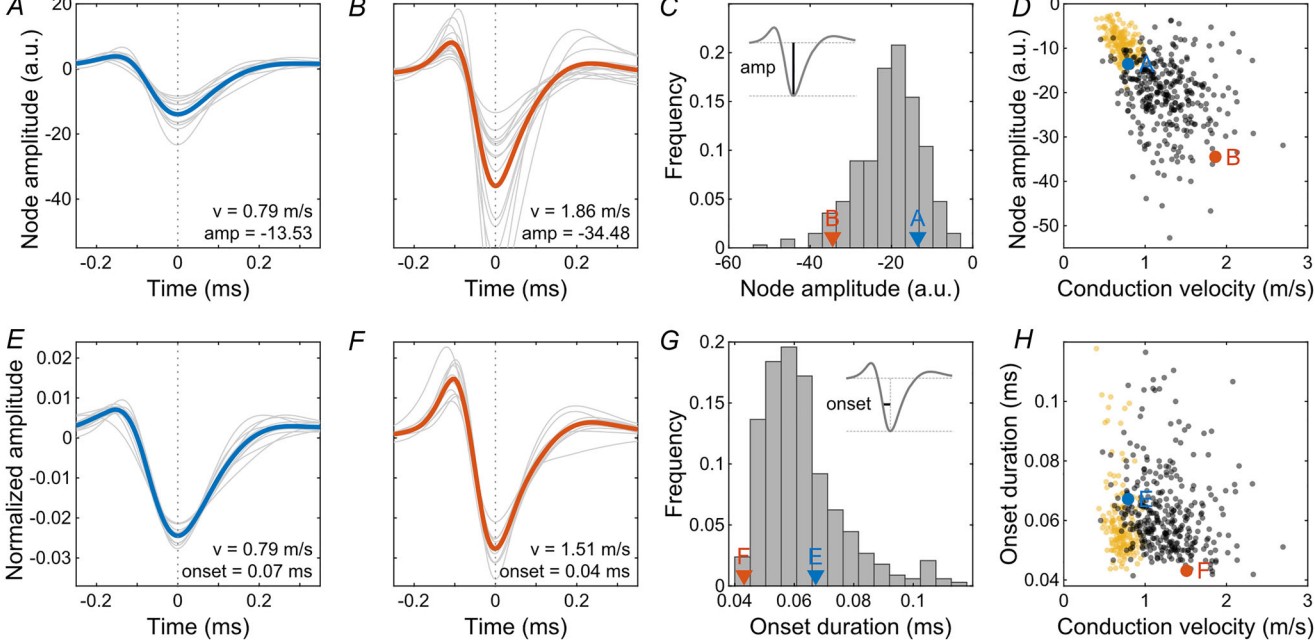

**Figure 4. The node amplitude and onset duration showed a large variation across axons in the European quail retina**

*A* and *B*, node amplitudes of saltatory axons of two RGCs. Slower (blue) and faster (red) axons marked for visual guidance as indicated in *C* and *D*. Grey lines represent individual nodes and the bold line represents their mean. *C*, histogram of robust mean node amplitudes across axons. RGCs of *A* and *B* marked. *D*, node amplitude against conduction velocity for saltatory axons (black). For comparison, the robust mean amplitudes of non-saltatory axons are shown (yellow). *E* and *F*, onset durations of saltatory axons of two RGCs. Grey lines represent individual nodes along one axon (normalized to absolute area) and the bold line represents their mean. *G*, histogram of robust mean onset durations across axons. RGCs of *E* and *F* marked. *H*, onset duration against conduction velocity for saltatory (black) and non-saltatory (yellow) axons.

Thus, those axons are not represented in Fig. 5*C*, as they were often negative for the neurofilament subunit. The expression levels of neurofilament differed between individual axons. Myelination and the corresponding clustering of sodium channels occurred at both strongly and weakly neurofilament-positive axons. The sodium channel cluster length was 1.4 ± 0.3 µm and the cluster width was 1.1 ± 0.3 µm ($n = 52$, 1 retina). Results of experiments with pigeon retina were consistent with those obtained from the quail, where panNa$_v$-immunoreactive nodes were located along neurofilament-positive axons (Fig. 5*D*).

The distances between sodium channel clusters were measured along strongly neurofilament-positive axons (Fig. 6*A*). This internode length (mean: 76.8 ± 25.2 µm, $n = 200$) was surprisingly variable along individual axons. However, the variability across axons was significantly larger (Fig. 6*B*; Wilcoxon rank-sum test: $P < 0.001$) suggesting that the internode lengths are cell type dependent. This is further supported by the finding that the internode length was positively correlated with the axon diameter (mean: 0.9 ± 0.2 µm, $n = 200$; correlation coefficient: 0.55, CI: 0.44, 0.64; $P < 0.001$; Fig. 6*C*), which was approximated as the width of the neurofilament labelling in the middle between each pair of nodes.

The internode length could be expected to be a major determinant of the axonal conduction velocity (Brill et al., 1977). However, qualitatively, this dependence appeared weak in our observations (Fig. 6*E–H*).

### Large interspecific variation in conduction velocities

We measured the conduction velocity in retinal axons of species with and without myelination, from different ecological niches, to explore potential adaptations (Fig. 7).

Notably, spike conduction velocities in the retinas of pigeon (2.07 ± 0.70 m/s; $n = 1135$) and zebra finch (1.55 ± 0.57 m/s, $n = 511$) were substantially higher than in those of the ground-dwelling European quail (1.29 ± 0.71 m/s; $n = 1032$) and Bankiva chicken (1.08 ± 0.41 m/s; $n = 1219$). In comparison, conduction velocities in unmyelinated mouse (0.65 ± 0.15 m/s; $n = 398$) and guinea pig (0.86 ± 0.18 m/s; $n = 2700$) retinas were on average slower than those of the avian species, in line with the absence of myelination (Kruskal–Wallis with Tukey–Kramer pairwise comparison tests: all combinations $P < 0.001$).

## Discussion

We examined the saltatory spike conduction of RGC axons and confirmed intraretinal myelination and the formation of nodes of Ranvier. On average, saltatory axons had higher conduction velocities than non-saltatory axons, although there was a notable overlap in the velocities between both groups. The observed differences in axonal conduction velocities between species seem to correspond to their respective ecological niches and behaviours.

The morphological features of the panNa$_v$-positive sodium channel clusters in axons of quail and pigeon retinas, including their tubular shape, size, distribution pattern along axons and gaps in the myelin sheath at those sites, closely resembled the structures observed in conventional nodes of Ranvier in the vertebrate optic nerve and other parts of the central nervous system (Fig. 5) (Arancibia-Cárcamo et al., 2017; Boiko et al., 2001; Caldwell et al., 2000; O'Brien et al., 2008; Rasband et al., 1999). We used a pan-antibody directed against a highly conserved sequence present in all vertebrate Nav1 isoforms (Rasband et al., 1999). Therefore, the exact sodium

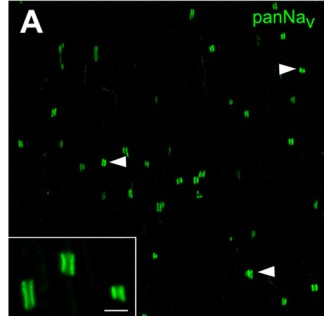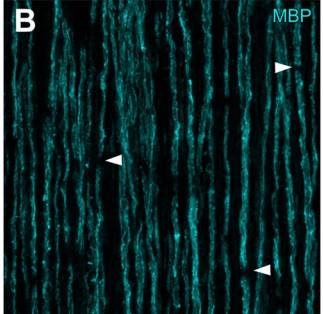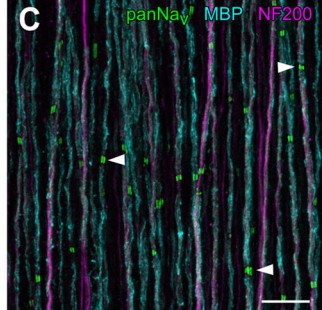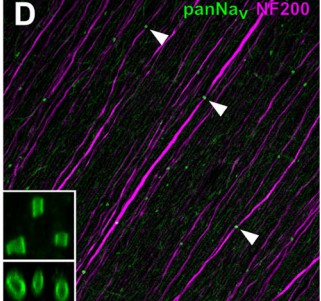

**Figure 5. Voltage-gated sodium channels were clustered at nodes of myelinated RGC axons in European quail and pigeon retinas**
*A–C*, confocal images of the nerve fibre layer in a quail retina immunolabeled against voltage-gated sodium channels (panNa$_v$), myelin basic protein (MBP), and neurofilament (NF200). Dense clusters of Na$_v$ channels were found along axons at sites where myelination was interrupted (example positions marked by arrowheads). Inset in *A* shows panNa$_v$ clusters at higher magnification. Scale bars: *A* inset: 2 µm, *C*: 10 µm. *D*, Na$_v$ channel clustering in the pigeon retina. As in the quail retina, clusters of Na$_v$ channels are found on axons (example positions marked by arrowheads). Inset shows panNa$_v$ clusters at higher magnification: Top: maximum intensity projection of an image stack. Bottom: side view of the rotated image stack. Scale bars: 50 µm, inset: 1 µm.

channel subunit expressed at nodes of Ranvier in quail and pigeon RGC axons remains unknown. In the mammalian retina, RGCs express Nav1.6 at the axon initial segment and Nav1.2 in their unmyelinated axons within the nerve fibre layer. There, nodes of Ranvier are formed only in the optic nerve, and Nav1.6 typically replaces Nav1.2 during development. Nav1.8 may also be present in alpha-like cell types with large-calibre axons (reviewed by Van Hook et al., 2019). Thus, it is tempting to speculate that, in birds, the diffuse panNav labelling observed in unmyelinated axons represents Nav1.2, while the clustered channels at nodes of Ranvier are formed mostly by Nav1.6, with Nav1.8 expressed only by a subset of particularly thick axons. However, further studies are required to identify the specific voltage-gated sodium channel subunits and their kinetics in RGC axons of the avian retina.

Overall, our functional results for the retina are consistent with data from other parts of the nervous system. However, specific constraints uniquely influence axonal conduction in the retina. The need for good optical transparency of the nerve fibre layer restricts the structure and amount of myelin present. Myelin acts as an electrical insulator that, by reducing the internodal membrane capacitance and membrane conductance, increases the length constant, decreases the time constant, and ultimately raises the conduction velocity of the

axon. Thus, it is likely that a balance exists between the conduction velocity and the preservation of the optical quality of the nerve fibre layer. In fact, only a subset of the intraretinal RGC axons are myelinated (Imagawa et al., 1999; Inoue et al., 1980). It is plausible that myelination and fast conduction velocities preferentially benefit RGC types transmitting information related to alertness or motion processing. Additionally, all centrifugal axons were described to be myelinated intraretinally (Wilson & Lindstrom, 2011).

While the highest conduction velocities were found only in saltatory axons and the lowest velocities in non-saltatory axons, we observed a surprisingly large overlap in velocities between saltatory and non-saltatory axons (Fig. 2). The conduction velocity of myelinated axons seems to benefit more from an increase in axon diameter (Brill et al., 1977; Waxman, 1980; see also Hodes, 1953; Hoffmeister et al., 1991; Pumphrey & Young, 1938) and myelinated axons are expected to reach higher velocities than unmyelinated axons for diameters larger than 0.2 μm (Waxman & Bennett, 1972). Additionally, higher velocities in unmyelinated axons might be achieved by an increased channel density (Schmidt & Knösche, 2019). Thus, the observed low velocities of saltatory, that is myelinated, axons could have been achieved by unmyelinated axons of comparable

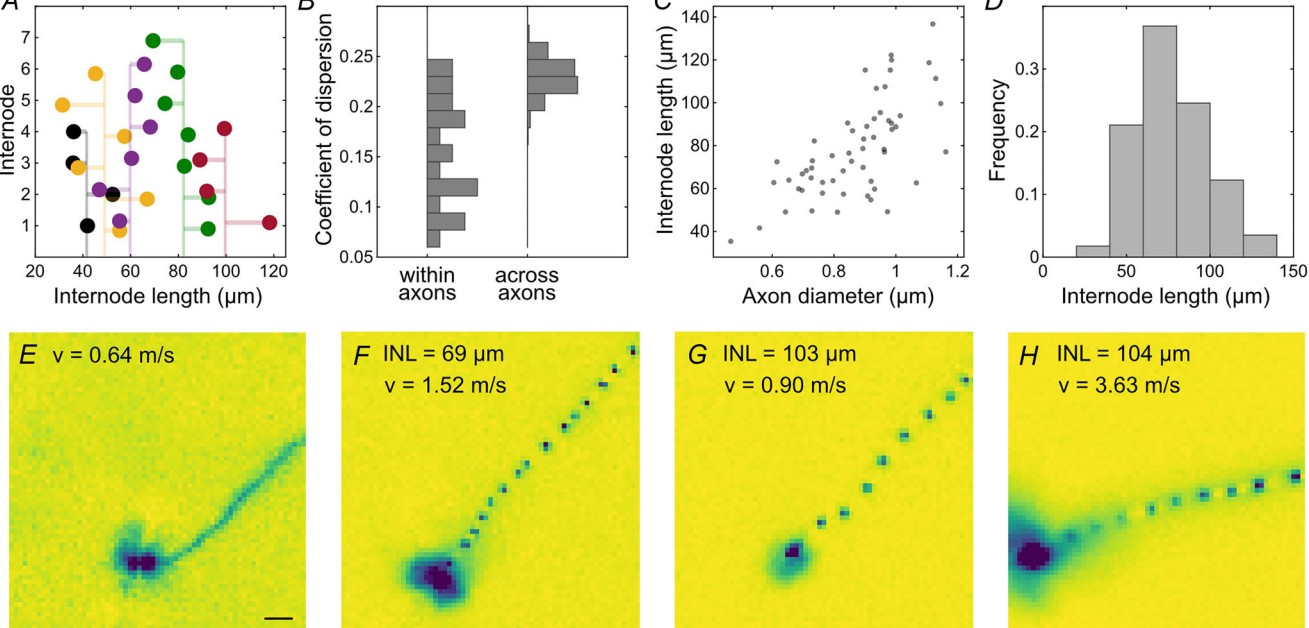

**Figure 6. The internode length showed a large variation across axons in the European quail retina**
*A*, successive internode lengths along five example axons. Vertical line represents the mean internode length for a given axon. *B*, the variation of internode lengths is significantly smaller within an axon than across axons (quartile coefficient of dispersion). Subselection of axons with at least five measured nodes (*n* = 22). Variance across axons is bootstrapped. *C*, the distances between nodes increase as a function of axon diameters. Mean for individual axons (*n* = 57). *D*, histogram of number of internode lengths. *E*–*H*, electrical images recorded with a MEA with electrode-to-electrode distance of 16 μm (65 × 65 electrodes). INL indicates the average internode length and *v* the conduction velocity. Examples from two retinas. Recorded at 36°C. Scale bar: 100 μm.

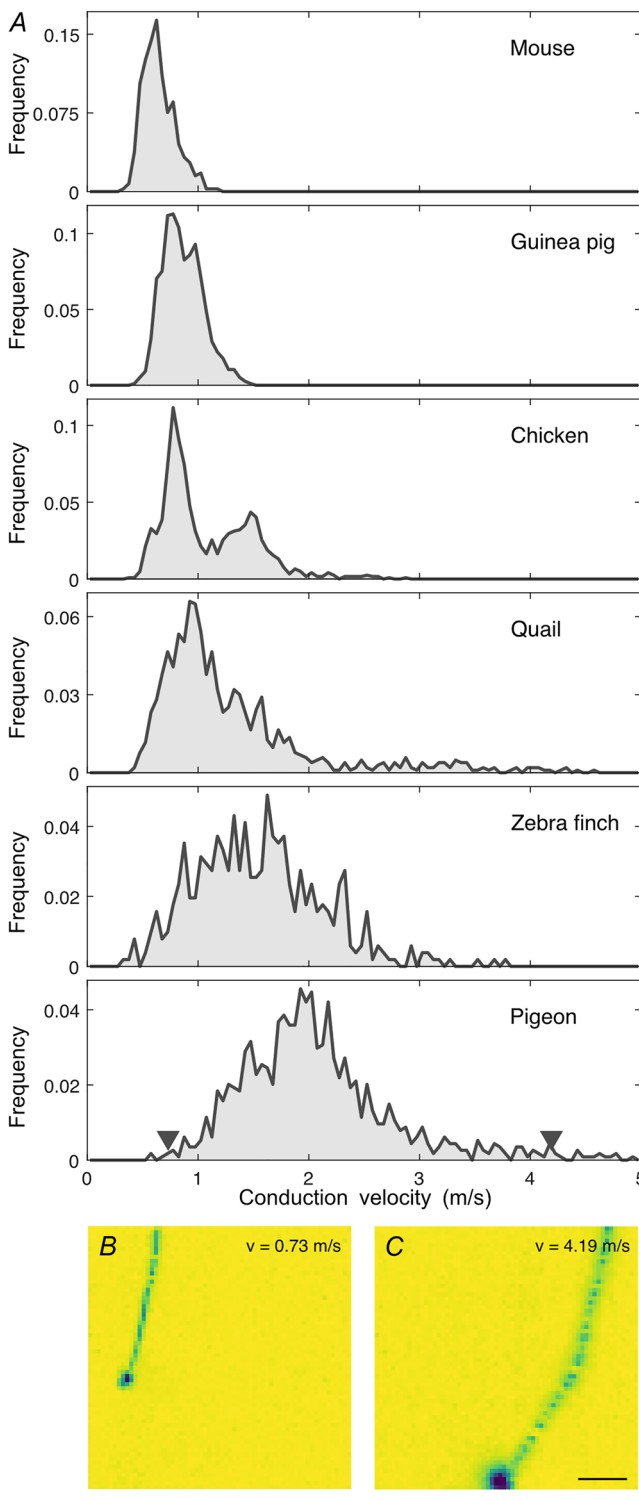

**Figure 7. Large interspecific diversity of conduction velocities**
*A*, axonal conduction velocity histograms for mouse (2 retinas, 398 axons), guinea pig (2 retinas, 2700 axons), European quail (2 retinas, 1032 axons), Bankiva chicken (2 retinas, 1219 axons), zebra finch (2 retinas, 511 axons) and pigeon (2 retinas, 1135 axons). Axons of insufficient length were excluded. Histograms were normalized to the total number of axons analysed for the respective species. Arrowheads in the last panel indicate velocities of cells shown in *B* and *C*. Note that the recordings were performed below body

temperature of the animals (see methods). *B* and *C*, example electrical images of slow (*B*) and fast (*C*) axonal spike conduction in pigeon RGCs. Scale bar: 500 μm. Soma amplitude saturated in *B* and *C* for visual clarity.

diameter, potentially providing better transparency. For a simple optimization, one might expect a clear boundary between the velocities of saltatory and non-saltatory axons. However, the significant overlap observed suggests a more intricate optimization process. For example, increasing the number of myelinated cell types might be constrained by the need for optical transparency, even as the information carried by additional cell types might require higher conduction velocities. Conversely, myelinated axons might offer additional benefits, but may derive limited benefit from further increases in velocity. A potential advantage of myelination could be an increased spatial packing density. However, at low conduction velocities, the total diameter of a myelinated axon including its sheath is generally larger than that of an unmyelinated axon (Waxman, 1980). Furthermore, while the metabolic advantage of myelination is particularly beneficial for axons with higher firing rates (Perge et al., 2009) we found no obvious difference in the firing rates of myelinated and unmyelinated axons of similar velocity, providing no evidence of a metabolic optimization. Nevertheless, myelination might entail other benefits for information transmission like, for example, reduced spike-timing jitter (Kim et al., 2013).

The measured intraretinal conduction velocities were largely consistent with recent findings in the chicken (Fig. 7) (Seifert et al., 2023). The intraretinal conduction velocities were relatively low compared to other parts of the central nervous system. For example, cortico-spinal neurons can reach conduction velocities of 100 m/s (cat, Takahashi, 1965), thalamic-cortical conduction velocities are in the range of 6–38 m/s (rabbit, Swadlow & Weyand, 1985), but also slow cortical neurons with velocities as low as 0.3 m/s have been described (rabbit, Swadlow, 1990). Extraretinal conduction velocities of RGC axons are indeed higher than intraretinal velocities (Stanford, 1987; Stone & Freeman, 1971), suggesting that a fast conduction is generally beneficial and further substantiating the notion of constrained intraretinal myelination.

While the variation of measured internode lengths along individual axons (Fig. 6*A*) might seem unexpected, similar variations have been described in regions of the peripheral and central nervous systems of different vertebrates (chicken auditory brainstem: Seidl et al., 2010; zebrafish spinal cord: Auer et al., 2018; Vagionitis et al., 2022; rabbit spinal cord: Hess & Young, 1949; rabbit peripheral nerves: Vizoso & Young, 1948). In our qualitative analysis, the conduction velocity appeared to have only a weak dependence on internode length (Fig. 6*E*–*H*).

Accordingly, theoretical studies predicted only a small effect of internode length on conduction velocity (Moore et al., 1978; Schmidt & Knösche, 2019).

A methodological bias concerns the temperature of the retinas during recordings. Although the temperature was relatively high by common experimental practice, it was below the body temperatures of the respective animals. Conduction velocity is expected to increase by roughly 50% over 10°C (reviewed in Waxman, 1980). Therefore, the conduction velocities should not be considered in absolute terms. The mismatch between body temperature and *ex vivo* recording temperature was largest for the avian species. In these cases our results might under-estimate the conduction velocities the most. Furthermore, the detection of nodes in the physiological data was limited by the electrode-to-electrode distance of the MEA. According to the sampling theorem, internode lengths below ∼90 μm could not be reliably detected and might have led to an under-representation of RGCs with smaller internode lengths in our analyses. However, control measurements with a higher resolution MEA (Fig. 6*E–H*) showed no obvious differences in the observed saltatory patterns. Conversely, the anatomical analysis was likely biased toward smaller internode lengths because RGC axons could only be reliably traced over relatively short distances based on the neurofilament staining. Another potential source of methodological bias could stem from the fact that all tested species were to varying degrees domesticated, which might have influenced their intra-retinal spike conduction velocities. For instance, pigeons have been primarily bred for orientation and flight speed, whereas zebra finches have been selected for their aesthetic plumage. In contrast, we recorded from the retinas of the Bankiva chicken, the wild ancestor of domesticated chickens, and the European quail, which is less commonly bred for poultry production compared to the Japanese quail.

We observed significantly larger conduction velocities in pigeon and zebra finch retinas than in those of European quail and Bankiva chicken. These interspecific differences in conduction velocities align with their respective ecological niches and behaviours, particularly the distinctions between ground-dwelling, arboreal and aerial lifestyles. The European quail for example is a pre-dominantly ground-dwelling bird species. While it travels long distances at night during the migratory season, it gathers the majority of food and builds nests near the ground. When threatened, it typically freezes and relies on camouflage, only taking flight as a last resort for short distances before returning to the ground and freezing again. On the other hand, pigeons and zebra finches are fast-flying birds requiring rapid visual processing (Boström et al., 2016; Potier et al., 2020). Their flocking behaviour necessitates quick reactions to directional changes by leading animals, as do flight manoeuvres near

obstacles such as perches. Moreover, they are far more likely to be attacked mid-flight by predators, where agile flight manoeuvres may provide a critical chance of escape. Furthermore, variations in eye size across species might contribute to differences in intraretinal delays. While there is evidence that spike conduction velocity within the retina can compensate for latency (Bucci et al., 2024), it remains unclear whether a similar optimization exists across species in relation to eye size. There appears to be a weak general trend suggesting that larger eyes are associated with higher conduction velocities. For example, in our data, the highest maximal conduction velocities were observed in the pigeon, and the lowest maximal velocities in the mouse. However, due to the large eyes of the pigeon and the offset position of its optic nerve, the resulting differences in spike arrival time were roughly equalized for the longest axons. At the same time, the guinea pig has larger eyes than the zebra finch, yet exhibits significantly slower maximum conduction velocities. If the pigeon had the maximal axonal conduction velocities of the mouse, the observed difference would correspond to a delay of ∼10 ms. For a fast flying pigeon this delay would correspond to a flight distance of 27 cm. The conduction delay difference between the slowest, non-saltatory axon and the fastest, saltatory axon in the pigeon retina was 24 ms, corresponding to a flight distance of 67 cm.

The avian retina is a striking example of adaptations in retinal physiology to the ecology of the species. Intra-retinal axonal conduction in birds seems to represent a multifaceted adaptation finely tuned to their ecological demands, balancing the requirements for speed, timing precision, metabolic efficiency and optical transparency.

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

## Additional information

### Data availability statement

The data is available on the University of Oldenburg Dataverse server: https://doi.org/10.57782/BWMQUV

## Competing interests

The authors declare they have no competing interests.

## Author contributions

C.P., M.G. and M.T.A. conceptualized the study. C.T.B., C.P., M.G. and M.T.A. provided the methodologies. C.T.B., C.P. and M.T.A. curated data for this study. C.T.B., M.T.A. and M.M. analysed the study. C.P., C.T.B., M.M., D.R.P. and M.G. acquired data. C.T.B., MG, M.M. and M.T.A. visualized the study. M.T.A. and M.G. wrote the original draft. C.T.B., C.P., D.R.P. and M.M. reviewed and edited the writing. C.P. and M.G. supervised the study. All authors have approved the final version of the manuscript and agreed to be accountable for all aspects of the work in ensuring that questions related to the accuracy or integrity of any part of the work are appropriately investigated and resolved. All persons designated as authors qualify for authorship, and all those who qualify for authorship are listed.

## Funding

This research was supported by SFB 1372 and RTG 1885/2, Deutsche Forschungsgemeinschaft (M.G.).

## Acknowledgements

We are grateful to Karin Dedek, Heiko Schmaljohann, Henrique von Gersdorff and Michael Winklhofer for their comments on an earlier version of the manuscript.

## Author's present address

C. Puller: Department of Computational Neuroethology, Max Planck Institute for Neurobiology of Behavior – caesar, Bonn, Germany.

## Keywords

Avian, multi electrode array, retinal ganglion cell, saltatory conduction

## Supporting information

Additional supporting information can be found online in the Supporting Information section at the end of the HTML view of the article. Supporting information files available:

**Peer Review History**

