## [Peer Review History · The Journal of Physiology]

Saltatory axonal conduction in the avian retina

Christoph Tobias Block, Malte Thorsten Ahlers, Christian Puller, Max Manackin, Dipti Ranjan Pradhan, and Martin Greschner

DOI: 10.1113/JP288664

Corresponding author(s): Malte Ahlers (m.ahlers@uol.de)

Review Timeline:

Submission Date:	30-Jan-2025
Editorial Decision:	29-Apr-2025
Revision Received:	11-Jul-2025
Editorial Decision:	21-Jul-2025
Revision Received:	22-Jul-2025
Accepted:	25-Jul-2025

Senior Editor: Nathan Schoppa

Reviewing Editor: Nathan Schoppa

Transaction Report:

Dear Dr Ahlers,

Re: JP-RP-2025-288664 "Saltatory axonal conduction in the avian retina" by Christoph Tobias Block, Malte Thorsten Ahlers, Christian Puller, Max Manackin, Dipti Ranjan Pradhan, and Martin Greschner

Thank you for submitting your manuscript to The Journal of Physiology. It has been assessed by a Reviewing Editor and by 1 expert referee and we are pleased to tell you that it is potentially acceptable for publication following satisfactory major revision.

Please address all the points raised and incorporate all requested revisions or explain in your Response to Referees why a change has not been made. We hope you will find the comments helpful and that you will be able to return your revised manuscript within 2 months. If you require longer than this, please contact journal staff: jp@physoc.org. Please note that this letter does not constitute a guarantee for acceptance of your revised manuscript.

REVISION CHECKLIST:

Please upload two versions of your manuscript text: one with all relevant changes highlighted and one clean version with no

changes tracked. The manuscript file should include all tables and figure legends, but each figure/graph should be uploaded as separate, high-resolution files.

We look forward to receiving your revised submission.

Yours sincerely,

Nathan Schoppa
Senior Editor
The Journal of Physiology

REQUIRED ITEMS

- Author photo and profile. First or joint first authors are asked to provide a short biography (no more than 100 words for one author or 150 words in total for joint first authors) and a portrait photograph. These should be uploaded and clearly labelled together in a Word document with the revised version of the manuscript. See Information for Authors for further details.

- The contact information for the person responsible for 'Research Governance' at your institution needs to be provided. This includes their name and an institutional email address. Please ensure the contact is not an author on this paper and provide an alternate contact if necessary, or confirm in the submission form that the author whose email was provided has sole responsibility for research governance. This is the person who is responsible for regulations, principles and standards of good practice in research carried out at the institution, for instance the ethical treatment of animals, the keeping of proper experimental records or the reporting of results.

- You must start the Methods section with a paragraph headed Ethical approval (https://jp.msubmit.net/cgi-bin/main.plex?form_type=display_requirements#methods).

Research must comply with The Journal's policies regarding animal experiments (<https://physoc.onlinelibrary.wiley.com/hub/animal-experiments>) and adherence to these policies must be stated in the manuscript.

Authors should confirm in their Methods section that their experiments were carried out according to the guidelines laid down by their institution's animal welfare committee, including an ethics approval reference number. The Methods section must contain a statement about access to food, water and housing, details of the anaesthetic regime: anaesthetic used, dose and route of administration, and method of killing the experimental animals.

- Your manuscript must include a complete Additional Information section, including competing interests; funding; author contributions and acknowledgements.

- Please upload separate high-quality figure files via the submission form.

- Please ensure that the Article File you upload is a Word file.

- Papers must comply with the Statistics Policy: https://jp.msubmit.net/cgi-bin/main.plex?form_type=display_requirements#statistics.

In summary:

- If $n \leq 30$, all data points must be plotted in the figure in a way that reveals their range and distribution. A bar graph with data points overlaid, a box and whisker plot or a violin plot (preferably with data points included) are acceptable formats.
 - If $n > 30$, then the entire raw dataset must be made available either as supporting information, or hosted on a not-for-profit repository, e.g. FigShare, with access details provided in the manuscript.
 - 'n' clearly defined (e.g. x cells from y slices in z animals) in the Methods. Authors should be mindful of pseudoreplication.
 - All relevant 'n' values must be clearly stated in the main text, figures and tables.
 - The most appropriate summary statistic (e.g. mean or median and standard deviation) must be used. Standard Error of the Mean (SEM) alone is not permitted.
 - Exact p values must be stated. Authors must not use 'greater than' or 'less than'. Exact p values must be stated to three significant figures even when 'no statistical significance' is claimed.
- Please include an Abstract Figure file, as well as the Figure Legend text within the main article file. The Abstract Figure is a piece of artwork designed to give readers an immediate understanding of the research and should summarise the main conclusions. If possible, the image should be easily 'readable' from left to right or top to bottom. It should show the physiological relevance of the manuscript so readers can assess the importance and content of its findings. Abstract Figures should not merely recapitulate other figures in the manuscript. Please try to keep the diagram as simple as possible and without superfluous information that may distract from the main conclusion(s). Abstract Figures must be provided by authors no later than the revised manuscript stage and should be uploaded as a separate file during online submission labelled as File Type 'Abstract Figure'. Please also ensure that you include the figure legend in the main article file. All Abstract Figures should be created using BioRender. Authors should use The Journal's premium BioRender account to export high-resolution images. Details on how to use and access the premium account are included as part of this email.

EDITOR COMMENTS

Reviewing Editor:

I agree with the assessment of the reviewer and support their recommendations for revision. Further, in the text, there should be clearer references to the subparts of Figures 3 and 4. In addition, the authors should state the body temperature of each animal under study in the methods section, so that the reader can appreciate the difference to the temperature used in vitro.

Please also see 'Required Items' above.

Senior Editor:

Comments for Authors to ensure the paper complies with the Statistics Policy (Required):

1. Conclusions based on differences in means and correlations are made without consistently providing information about statistical tests or p-values. Information about statistics should be provided in every instance in which comparisons or correlations are made.

2. N values should be consistently provided in the main text when comparisons are made.

Comments to the Author:

Thank you for submitting your manuscript to Journal of Physiology (JP). It has been reviewed by one external referee and a reviewing editor (RE), who felt that the study is addressing an interesting issue and that the results are novel. A concern of the referee was the use of non-physiological temperatures but they (and the JP editorial staff) believe that the issue is addressed well in the text and the results still potentially important. A number of concerns however were raised by both the external referee and RE that will need to be addressed with changes in the text and additional analyses.

I would also amplify one of the concerns raised by the referee about the need for additional statistical analysis. They raised the concern around the correlations discussed around Fig. 6E-H but the description of the statistical analysis throughout the study needs much improvement. Many statements appear to be made based on just mean values and standard deviations or values for the correlation coefficient with no information about statistical tests performed or p-values. An example would be around line 293: "As expected, saltatory axons displayed overall higher conduction velocities (saltatory: 1.25 {plus minus} 0.35 m/s, non-saltatory: 0.71 {plus minus} 0.12 m/s)."

Information about statistics (test and p-value) needs to be provided in every instance in which results are described at the point at which they are described. N values should also be consistently provided in the main text and figure legends. There is some discussion of statistics in the methods but it appears to be specific to only a few points and not general. The authors should follow all guidelines about presentation of data and statistics provided by JP.

REFeree COMMENTS

Referee #2:

The manuscript by Block et al. describes measurements of avian and rodent retinae using microelectrode arrays. In particular, they investigate the action potential conduction velocity of the axons of retinal ganglion cells. They report signatures of saltatory conduction in the extracellular waveforms of the action potentials in avian retina and describe the relationship of conduction velocity with other physiological parameters, like internode distance.

The paper is generally well-written and well-structured, the results are novel, and the characterization is convincing.

My biggest concern (which is not that big) with the results, and which somewhat limits their generalizability, is the use of unphysiological conditions to measure conduction velocity. The authors are very clear about this limitation, though. However, I believe a comparison of conduction speeds at physiologically relevant body temperatures would have been more meaningful. Also, a statement discussing differences (or the lack thereof) in eye sizes could enhance the discussion of the impact of conduction delays.

I have no major concerns with the paper, only minor suggestions that the authors may consider at their discretion.

Minor:

-The use of "their" in the second sentence of the abstract technically refers to "parts" of the first, which may cause ambiguity.

-L36: "It can be"-the authors could give a bit more credit to others by stating that this was previously hypothesized.

-Although I am not a native English speaker myself, some sentences sounded slightly unnatural. For example, L39: "imaged ... conduction velocity ... in comparison to" is phrased awkwardly. The presentation of the paper could likely benefit from improvements in language clarity and explanations (see also below).

-"Conversely, the internode length and the time required for a node to activate were weak predictors of conduction velocity." Just reading the abstract, this sentence is hard to understand, as the temporal reference is unclear. "Time required"-with respect to what time point? This phrasing seems contradictory to the preceding sentence.

-L73: Extraneous ")" character.

-L183-185: Explanation is difficult to parse. What does "after averaging" in L184 refer to? Clarifying this point with additional explanation would be helpful.

-L393ff: "However, no strong correlation between the internode length and

conduction velocity was apparent in our data (Figure 6 E-H). Axons with similar internode lengths exhibited vastly different conduction velocities and axons with similar conduction

velocities had different internode lengths." These statements could be strengthened by additional statistical analysis and scatter plots. This is a surprising result-since conduction velocity is expected to correlate with axon diameter, and internodal length appears to correlate with diameter, one would also expect internodal length to correlate with velocity. Some additional discussion or data could clarify this point.

-The authors may wish to cite a recent preprint (<https://www.biorxiv.org/content/10.1101/2024.04.30.591867v1>) on conduction velocities in primate retina, as it employs similar techniques.

-Several instances of extra spaces " " (e.g., L516) should be corrected.

END OF COMMENTS

Response to Referees

We thank the reviewer and editors for taking the time to evaluate our manuscript and for their constructive comments. We have addressed the points raised. Detailed responses are provided below, and substantial revisions are marked in the manuscript.

Comments to the Author:

Thank you for submitting your manuscript to Journal of Physiology (JP). It has been reviewed by one external referee and a reviewing editor (RE), who felt that the study is addressing an interesting issue and that the results are novel. A concern of the referee was the use of non-physiological temperatures but they (and the JP editorial staff) believe that the issue is addressed well in the text and the results still potentially important. A number of concerns however were raised by both the external referee and RE that will need to be addressed with changes in the text and additional analyses.

I would also amplify one of the concerns raised by the referee about the need for additional statistical analysis. They raised the concern around the correlations discussed around Fig. 6E-H but the description of the statistical analysis throughout the study needs much improvement. Many statements appear to be made based on just mean values and standard deviations or values for the correlation coefficient with no information about statistical tests performed or p-values. An example would be around line 293: "As expected, saltatory axons displayed overall higher conduction velocities (saltatory: 1.25 {plus minus} 0.35 m/s, non-saltatory: 0.71 {plus minus} 0.12 m/s)."

Information about statistics (test and p-value) needs to be provided in every instance in which results are described at the point at which they are described. N values should also be consistently provided in the main text and figure legends. There is some discussion of statistics in the methods but it appears to be specific to only a few points and not general. The authors should follow all guidelines about presentation of data and statistics provided by JP.

We adjusted the manuscript to address the raised concerns and provided the statistical information consistently throughout the main text and figure legends.

Reviewing Editor:

I agree with the assessment of the reviewer and support their recommendations for revision. Further, in the text, there should be clearer references to the subparts of Figures 3 and 4.

Thank you, we improved the section to clarify the references to the subpanels.

In addition, the authors should state the body temperature of each animal under study in the methods section, so that the reader can appreciate the difference to the temperature used in vitro.

We added an indication of the body temperature of all animals under study (L 139 ff).

Please also see 'Required Items' above.

We addressed all points.

Senior Editor:

Comments for Authors to ensure the paper complies with the Statistics Policy (Required):

1. Conclusions based on differences in means and correlations are made without consistently providing information about statistical tests or p-values. Information about statistics should be provided in every instance in which comparisons or correlations are made.

We provided the statistical information consistently throughout the main text.

2. N values should be consistently provided in the main text when comparisons are made.

We included the n-values accordingly.

Referee #2:

The manuscript by Block et al. describes measurements of avian and rodent retinae using microelectrode arrays. In particular, they investigate the action potential conduction velocity of the axons of retinal ganglion cells. They report signatures of saltatory conduction in the extracellular waveforms of the action potentials in avian retina and describe the relationship of conduction velocity with other physiological parameters, like internode distance.

The paper is generally well-written and well-structured, the results are novel, and the characterization is convincing.

My biggest concern (which is not that big) with the results, and which somewhat limits their generalizability, is the use of unphysiological conditions to measure conduction velocity. The authors are very clear about this limitation, though. However, I believe a comparison of conduction speeds at physiologically relevant body temperatures would have been more meaningful. Also, a statement discussing differences (or the lack thereof) in eye sizes could enhance the discussion of the impact of conduction delays.

I have no major concerns with the paper, only minor suggestions that the authors may consider at their discretion.

We thank the reviewer for this assessment. We agree that the temperature variations are unfortunate; not all recordings were collected with this specific study in mind. We have added a discussion of the relationship between eye size and conduction velocities (L 580 ff and L 229 ff).

Minor:

-The use of "their" in the second sentence of the abstract technically refers to "parts" of the first, which may cause ambiguity.

Thank you - reformulated

-L36: "It can be"-the authors could give a bit more credit to others by stating that this was

previously hypothesized.

Thank you - reformulated

-Although I am not a native English speaker myself, some sentences sounded slightly unnatural. For example, L39: "imaged ... conduction velocity ... in comparison to" is phrased awkwardly. The presentation of the paper could likely benefit from improvements in language clarity and explanations (see also below).

Thank you - reformulated

-"Conversely, the internode length and the time required for a node to activate were weak predictors of conduction velocity." Just reading the abstract, this sentence is hard to understand, as the temporal reference is unclear. "Time required"-with respect to what time point? This phrasing seems contradictory to the preceding sentence.

Thank you - reformulated

-L73: Extraneous ")" character.

Corrected

-L183-185: Explanation is difficult to parse. What does "after averaging" in L184 refer to? Clarifying this point with additional explanation would be helpful.

We rewrote the paragraph to improve the readability (L 212 ff).

-L393ff: "However, no strong correlation between the internode length and conduction velocity was apparent in our data (Figure 6 E-H). Axons with similar internode lengths exhibited vastly different conduction velocities and axons with similar conduction velocities had different internode lengths." These statements could be strengthened by additional statistical analysis and scatter plots. This is a surprising result-since conduction velocity is expected to correlate with axon diameter, and internodal length appears to correlate with diameter, one would also expect internodal length to correlate with velocity. Some additional discussion or data could clarify this point.

Most recordings were and are performed using arrays with a 42 μm electrode pitch, which potentially undersample small internode distances (see Reviewer figure). Therefore, we prefer to omit this data from the paper and present this section qualitatively, using example cells recorded with a 16 μm pitch array (Fig. 6E–H). We reformulated our findings to reflect this (L 427 f).

Reviewer figure: Internode lengths based on recordings with an electrode-to-electrode distance of 42 μm . **A:** Histogram of number of internode lengths. Internode length is measured as average center to center distance. **B:** Internode length against conduction velocity.

-The authors may wish to cite a recent preprint (<https://www.biorxiv.org/content/10.1101/2024.04.30.591867v1>) on conduction velocities in primate retina, as it employs similar techniques.

Thank you for the pointer - we added the reference.

-Several instances of extra spaces " " (e.g., L516) should be corrected.

Corrected

Dear Dr Ahlers,

Re: JP-RP-2025-288664R1 "Saltatory axonal conduction in the avian retina" by Christoph Tobias Block, Malte Thorsten Ahlers, Christian Puller, Max Manackin, Dipti Ranjan Pradhan, and Martin Greschner

Thank you for submitting your manuscript to The Journal of Physiology. It has been assessed by a Reviewing Editor and we are pleased to tell you that it is acceptable for publication following satisfactory minor revision.

The review comments are copied at the end of this email.

REVISION CHECKLIST:

We look forward to receiving your revised submission.

Yours sincerely,

Nathan Schoppa
Senior Editor
The Journal of Physiology

REQUIRED ITEMS

- Papers must comply with the Statistics Policy: https://jp.msubmit.net/cgi-bin/main.plex?form_type=display_requirements#statistics.

In summary:

- If $n \leq 30$, all data points must be plotted in the figure in a way that reveals their range and distribution. A bar graph with data points overlaid, a box and whisker plot or a violin plot (preferably with data points included) are acceptable formats.
- If $n > 30$, then the entire raw dataset must be made available either as supporting information, or hosted on a not-for-profit repository, e.g. FigShare, with access details provided in the manuscript.
- 'n' clearly defined (e.g. x cells from y slices in z animals) in the Methods. Authors should be mindful of pseudoreplication.
- All relevant 'n' values must be clearly stated in the main text, figures and tables.
- The most appropriate summary statistic (e.g. mean or median and standard deviation) must be used. Standard Error of the Mean (SEM) alone is not permitted.
- Exact p values must be stated. Authors must not use 'greater than' or 'less than'. Exact p values must be stated to three significant figures even when 'no statistical significance' is claimed.

EDITOR COMMENTS

Reviewing Editor:

The authors have addressed the concerns raised in the first review. Congratulations on an interesting study!

Senior Editor:

Comments for Authors to ensure the paper complies with the Statistics Policy (Required):
For many of the results, p-values are provided with values indicated as "0.000". This presentation does not satisfy the requirement to report p-values to three significant figures, not three decimal places. Alternatively -- and I would say preferably in these cases -- they should indicate $p < 0.001$.

The journal's policy is stated here:

"For a given conclusion to be assessed, the exact p values must be stated to three significant figures (not decimal places)

even when 'no statistical significance' is being reported (i.e. for anything >0.001 , please report to 3 significant figures, e.g. 0.00236 or 0.523, etc.). These should be stated in the main text, figures and their legends and tables. The only exception to this is if p is less than 0.001, in which case '<' is permitted."

Comments to the Author:

Thank you for the revised manuscript. Your additions and revisions have addressed nearly all of the prior concerns adequately and we appreciate the additional information about statistics. The one exception is in how many of the p -values are now reported. For many of the results, p -values are provided with values indicated as "0.000". This presentation does not satisfy the requirement to report p -values to three significant figures, not three decimal places. Alternatively -- and I would say preferably in these cases -- the authors should indicate $p < 0.001$.

END OF COMMENTS

Response to Referees

We would like to thank you for accepting our manuscript for publication. We have adjusted the reporting of p-values in accordance with the journal's statistical policy. Specifically, p-values previously reported as $p = 0.000$ are now reported as $p < 0.001$.

Dear Dr Ahlers,

Re: JP-RP-2025-288664R2 "Saltatory axonal conduction in the avian retina" by Christoph Tobias Block, Malte Thorsten Ahlers, Christian Puller, Max Manackin, Dipti Ranjan Pradhan, and Martin Greschner

We are pleased to tell you that your paper has been accepted for publication in The Journal of Physiology.

Yours sincerely,

Nathan Schoppa
Senior Editor
The Journal of Physiology

If you would like to receive our 'Research Roundup', a monthly newsletter highlighting the cutting-edge research published in The Physiological Society's family of journals (The Journal of Physiology, Experimental Physiology, Physiological Reports, The Journal of Nutritional Physiology and The Journal of Precision Medicine: Health and Disease), please click this link, fill in your name and email address and select 'Research Roundup':

<https://www.physoc.org/journals-and-media/membernews>

- **TRANSPARENT PEER REVIEW POLICY:** To improve the transparency of its peer review process, The Journal of Physiology publishes online as supporting information the peer review history of all articles accepted for publication. Readers will have access to decision letters, including Editors' comments and referee reports, for each version of the manuscript as well as any author responses to peer review comments. Referees can decide whether or not they wish to be named on the peer review history document.
- You can help your research get the attention it deserves! Check out Wiley's free Promotion Guide for best-practice recommendations for promoting your work at: www.wileyauthors.com/eeo/guide. You can learn more about Wiley Editing Services which offers professional video, design, and writing services to create shareable video abstracts, infographics, conference posters, lay summaries, and research news stories for your research at: www.wileyauthors.com/eeo/promotion.
- **IMPORTANT NOTICE ABOUT OPEN ACCESS:** To assist authors whose funding agencies mandate public access to published research findings sooner than 12 months after publication, The Journal of Physiology allows authors to pay an Open Access (OA) fee to have their papers made freely available immediately on publication.

EDITOR COMMENTS

We appreciate the attention to the final details about p-values. Congratulations! The paper is now acceptable for publication.